# Interdependence between SEB-3 receptor and NLP-49 peptides shifts across predator-induced defensive behavioral modes in *Caenorhabditis elegans*

**Kathleen T Quach\*, Gillian A Hughes, Sreekanth H Chalasani\***

Molecular Neurobiology Laboratory, The Salk Institute for Biological Studies, La Jolla, United States

## eLife assessment

This study presents a **valuable** finding on predator threat detection in *C. elegans* and the role of neuropeptide systems in defensive behavioral strategies. The evidence supporting the conclusions is **solid**, although additional analyses and control experiments would strengthen the claims of the study. Overall, the work is of interest to the *C. elegans* community as well as neuroethologists and ecologists studying predator-prey interactions.

**\*For correspondence:**
kthln.t.qch@gmail.com (KTQ);
schalasani@salk.edu (SHC)

**Competing interest:** The authors declare that no competing interests exist.

**Abstract** Prey must balance predator avoidance with feeding, a central dilemma in prey refuge theory. Additionally, prey must assess predatory imminence—how close threats are in space and time. Predatory imminence theory classifies defensive behaviors into three defense modes: pre-encounter, post-encounter, and circa-strike, corresponding to increasing levels of threat——suspecting, detecting, and contacting a predator. Although predatory risk often varies in spatial distribution and imminence, how these factors intersect to influence defensive behaviors is poorly understood. Integrating these factors into a naturalistic environment enables comprehensive analysis of multiple defense modes in consistent conditions. Here, we combine prey refuge and predatory imminence theories to develop a model system of nematode defensive behaviors, with *Caenorhabditis elegans* as prey and *Pristionchus pacificus* as predator. In a foraging environment comprised of a food-rich, high-risk patch and a food-poor, low-risk refuge, *C. elegans* innately exhibits circa-strike behaviors. With experience, it learns post- and pre-encounter behaviors that proactively anticipate threats. These defense modes intensify with predator lethality, with only life-threatening predators capable of eliciting all three modes. SEB-3 receptors and NLP-49 peptides, key stress regulators, vary in their impact and interdependence across defense modes. Overall, our model system reveals fine-grained insights into how stress-related signaling regulates defensive behaviors.

## Introduction

To survive, prey adjust their behavior to avoid predatory threat across a variety of situations. This repertoire of defensive behaviors includes reactions to predatory attacks, as well as proactive behaviors that promote vigilance and reduce vulnerability. Due to strong selective pressure, many prey species evolved escape responses to rapidly evade predatory attacks (*Eaton, 2013*; *Evans et al., 2019*). While instinctive and not requiring conscious thought, these responses consume significant energy and are less effective against anticipated threats. In situations that do not demand immediate action, prey can flexibly adjust their defensive strategy based on the specific threat context. These defensive

strategies differ according to predatory imminence, which is the perceived spatial and temporal proximity of a predatory threat (*Fanselow and Lester, 1988*). In predatory imminence theory, defensive behaviors are categorized into three defense modes —pre-encounter, post-encounter, and circa-strike modes—each corresponding to increasing levels of predatory imminence—suspecting a predator, detecting a predator, and contact with a predatory attack (*Fanselow and Lester, 1988*). This framework has been primarily used to study rodent behaviors, such as escape actions (circa-strike), freezing (post-encounter), and altered meal patterns (pre-encounter), each linked to specific brain regions (*Fanselow and Lester, 1988*; *Fanselow et al., 1988*). Despite debates over relating animal defensive behaviors to human emotions like fear (*Mobbs et al., 2019*), the predatory imminence framework links circa-strike, post-encounter, and pre-encounter modes with panic, fear, and anxiety, respectively, based on threat and behavioral criteria rather than on similarity to humans in fear-related brain regions or responses to anxiolytic drugs (*Perusini and Fanselow, 2015*). However, this framework has predominantly been investigated with laboratory tests that use electric shocks (*Fanselow and Lester, 1988*; *Fanselow, 1989*; *Helmstetter and Fanselow, 1993*), rather than more naturalistic threat stimuli. Moreover, despite its species-agnostic approach, the predatory imminence framework has seldom been used to explore defensive behaviors in invertebrate models.

In naturalistic environments, prey face the dilemma of balancing the risk of predation with the need to forage, a challenge that intensifies in food-rich areas also marked by high predation risk. To navigate this balance, prey develop strategies to move between food-rich, high-risk areas and refuges, which are areas with less food but also low predation risk (*Sih, 1987*). Maximally safe strategies, such residing only in refuges, are often unsustainable as they result in starvation when food is scarce. Prey refuge theory, a branch of optimal foraging theory, identifies key factors influencing the use of refuges, such as predation risk, hunger level, feeding rates, and uncertainty (*Sih, 1992*). These factors are critical in post- and pre-encounter modes for assessing spaces for safety and adapting defensive strategies based on the particular risks and resources of the environment. Although previous studies, like those exploring the impact of electric shock on mice foraging in an operant chamber (*Fanselow et al., 1988*), have touched on these concepts, there has been little systematic integration of actual predators and refuges in lab studies guided by the predatory imminence framework. Recent approaches to evaluating defense modes across the predatory imminence spectrum rely on a battery of established laboratory tests (*Hoffman et al., 2022*), potentially complicating comparisons across defense modes due to widely varying experimental setups. Integrating predatory imminence and prey refuge theories enables us to develop behavioral tests for defense modes in a consistent and naturalistic environment, thus minimizing variables from different experimental designs.

Investigation of distinct defense modes within the same study can potentially shed light on the molecular regulation of threat behaviors and enhance the translatability of these insights. Since corticotropin-releasing factor (CRF) was identified in 1981 (*Vale et al., 1981*), its role in stress responses in both humans and animals has been a focus of research (*Bale and Vale, 2004*; *Binder and Nemeroff, 2010*), linking CRF system dysregulation to depression and anxiety, especially via the CRFR1 receptor (*Reul and Holsboer, 2002*; *Arborelius et al., 1999*; *Heinrichs et al., 1997*). However, CRFR1 antagonists, despite showing promise in animal studies, have struggled to become effective treatments in humans, partly because therapeutic indication is difficult to determine based on preclinical studies (*Spierling and Zorrilla, 2017*). These models often do not exhibit effects under normal conditions, requiring specific conditions to mimic stress responses, and the influence of CRF varies with the stress condition (*Zorrilla and Koob, 2004*). For instance, while high CRF levels correlate with PTSD in some human studies (*Bremner et al., 1997*; *Sautter et al., 2003*; *Baker et al., 1999*), this is not consistently seen in other anxiety disorders (*Banki et al., 1992*; *Fossey et al., 1996*; *Jolkkonen et al., 1993*). Additionally, research in mice shows that CRF can trigger opposite responses based on stress intensity (*Lemos et al., 2012*), suggesting the context of threat significantly impacts molecular mechanisms. However, variations in experimental setups and outcomes across studies complicate cross-study comparisons (*Bale and Vale, 2004*; *Atli et al., 2016*). Thus, to better understand the molecular dynamics of defensive behavior shifts, it is essential to study these behaviors through well-defined threat stages within a consistent framework in a single study.

To bridge this gap, we introduce a model system of nematode defensive behaviors with *Caenorhabditis elegans* as the prey and *Pristionchus pacificus* as the predator. Utilizing an invertebrate prey allows for investigation of interactions with life-threatening predators in a lab setting, avoiding

the ethical constraints faced by rodent research. The lack of life-endangering threats in vertebrate research has been criticized as a limitation in the translatability of rodent anxiety behavior tests (*Bach, 2022*). While *C. elegans* is an obligate bacteriovore, *P. pacificus* is omnivorous and can choose to eat bacteria, which it prefers, or to bite and kill nematode prey for food (*Serobyan et al., 2014*; *Wilecki et al., 2015*). *C. elegans* has been found alongside *Pristionchus* sp. nematodes in samples collected from the wild (*Félix et al., 2018*), suggesting that *C. elegans* may be more likely to recognize *P. pacificus* as a predator than other known artificial aversive stimuli, such as blue light or electric shocks. While *P. pacificus* can kill larval *C. elegans*, adults can survive hours of repeated biting (*Wilecki et al., 2015*; *Quach and Chalasani, 2022*), enabling them to learn from these encounters and adapt their behaviors. Additionally, *C. elegans* has been shown to form a learned association of a bacterial patch with predation risk, as *C. elegans* does not innately avoid food patches occupied by *Pristionchus* sp. or conditioned with their secretions (*Quach and Chalasani, 2022*; *Pribadi et al., 2023*). *P. pacificus* tends to stay within bacterial food patches (*Quach and Chalasani, 2022*), creating a natural setup of risky patches and safe refuge surrounding the patch. Leveraging this setup, our model system of nematode defensive behaviors applies predatory imminence and prey refuge theories to explore *C. elegans*' navigation of patch and refuge areas across defense modes.

Just as specific brain regions in rodents correlate with defense modes in predatory imminence theory, we aim to identify distinct molecular mechanisms driving defense modes in nematodes. Our focus is on SEB-3, a G protein-coupled receptor in *C. elegans* (*Jee et al., 2013*), and NLP-49, a neuropeptide locus where one of the peptides has been identified as a ligand for SEB-3 (*Beets et al., 2023*; *Chew et al., 2018*). Although SEB-3 initially appeared similar to mammalian CRF receptors, particularly CRFR1 (*Cardoso et al., 2006*; *Jee et al., 2013*; *Jee et al., 2016*), recent reports suggest that it is more closely related to invertebrate pigment-dispersing factor (PDF) receptors (*Elphick et al., 2018*; *Mirabeau and Joly, 2013*). Despite this, both CRF and PDF receptors are part of the secretin superfamily of receptors, with evidence suggesting that SEB-3 may influence some behaviors similarly to CRF receptors (*Jee et al., 2013*; *Jee et al., 2016*; *Chew et al., 2018*). Similar to the mammalian CRF signaling system, there are conflicting reports on the role SEB-3 signaling in nematode stress responses. The role SEB-3/NLP-49–3 signaling in stress response is debated, with some studies linking reduced signaling to low stress and increased signaling to high stress (*Jee et al., 2013*; *Chew et al., 2018*). One study indicating increased SEB-3 signaling reduces stress-like behaviors (*Jee et al., 2016*). This conflicting study differs from the others, which focus on basal stress indicators such as locomotion and arousal, by focusing on a choice between continuing to mate or escaping aversive blue light (*Jee et al., 2016*). Our model system of defensive behaviors also involves choosing between continuing an appealing activity and avoiding an aversive stimulus. Thus we hypothesize that decreased SEB-3/NLP-49-2 signaling will enhance defensive behaviors in our model system, while increased signaling will reduced defensive behaviors. However, we expect that the specific roles of and interactions between SEB-3 and NLP-49-2 will differ across defense modes. Overall, we demonstrate that our model system of nematode defensive behaviors can be successfully used to interrogate the specific behavioral targets of NLP-49 and SEB-3 signaling and interaction. By maintaining a consistent test environment across defense modes, we are able to attribute differences in molecular regulation to the defense mode itself, facilitating a more robust understanding of stress-related molecular signaling.

## Results

### *Caenorhabditis elegans* responses to predatory threat can be organized into three defense modes

To focus our model system of nematode defensive behaviors around a bacterial food patch and refuge, we adapted our previous predator–prey competition model (*Quach and Chalasani, 2022*) to concentrate on the behavior of the prey rather than that of the predator. This system examines interactions among three species across different trophic levels: (1) *C. elegans* as prey, (2) *P. pacificus* as the predator, and (3) a localized food source (patch) of OP50 *Escherichia coli* bacteria (*Figure 1A*). *P. pacificus* is territorial over small patches of bacterial food, such that it resides mostly within the patch and patrols the patch border for intruders (*Quach and Chalasani, 2022*). This results in *C. elegans* experiencing predatory attacks (bites) mostly when it contacts the patch, especially at the patch boundary, such that predation risk is primarily confined to the patch. In contrast, the surrounding refuge area

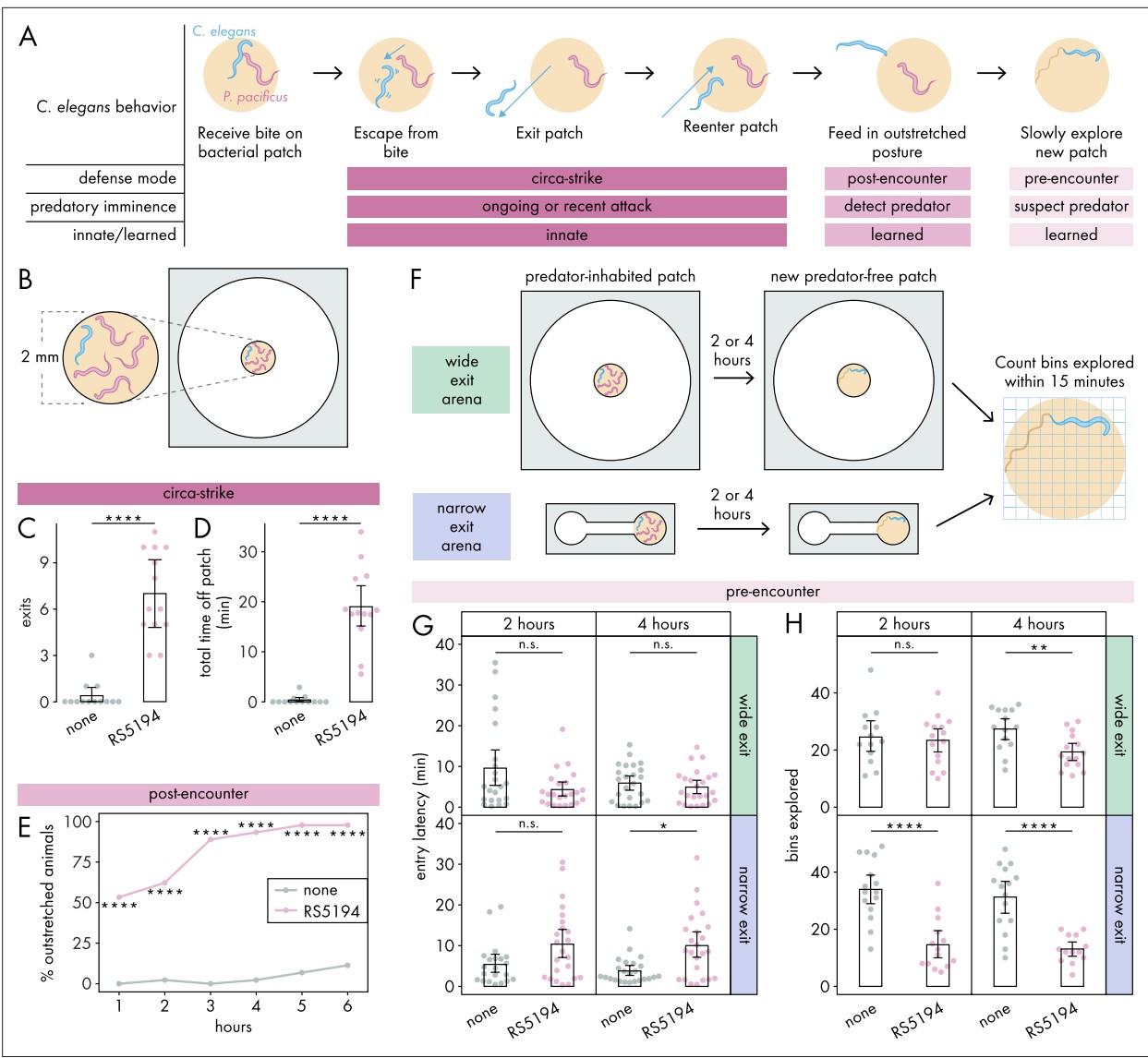

**Figure 1.** *C. elegans* responses to predatory threat can be organized into three defense modes. (**A**) Predatory imminence model of *C. elegans* defensive responses to a predator-inhabited bacterial food patch. Upon being bitten by a predator, *C. elegans* executes an escape response, exits the predator-inhabited patch, and then ultimately reenters the patch (circa-strike mode). After extended exposure, *C. elegans* adopts an outstretched feeding posture to minimize predator contact (post-encounter mode). When confronted with a new patch, the predator-exposed *C. elegans* cautiously explores for potential predators in the patch (pre-encounter mode). (**B**) Arena setup for assessing circa-strike and post-encounter behaviors includes a 9.5 mm circular arena with a 2 mm bacterial patch, housing one *C. elegans* and four RS5194 *P. pacificus* predators (or none). (**C**) Number of exits and (**D**) total time that *C. elegans* spent off the patch during 1 hour exposure to predator and predator-free conditions (Dunn's test, $n_{C.\ elegans}$ = 13). (**E**) Percentage of *C. elegans* animals adopting outstretched feeding posture across different exposure durations to both predator and predator-free conditions (Fisher's exact test, $n_{C.\ elegans}$ = 44–45). (**F**) Arena setup for studying pre-encounter behaviors involved placing one *C. elegans* and four RS5194 *P. pacificus* predators (or none) in either a wide exit arena (open space around the food patch) or a narrow exit arena (narrow corridor to/from the food patch) with a 2 mm bacterial patch. After 2- or 4-hour predator exposure, *C. elegans* was transferred to a predator-free arena for a 15-minute exploration period. (**G**) Latency to enter a new patch (Dunn's test with Benjamini–Hochberg adjustment, $n_{C.\ elegans}$ = 21–24) and (**H**) number of bins explored by *C. elegans* following either 2- or 4-hour exposure to predator or predator-free conditions (Welch's *t*-test with Benjamini–Hochberg adjustment, $n_{C.\ elegans}$ = 13–15). Error bars are 95% bootstrap CIs containing the mean. n.s. = p>0.05, *p<0.05, **p<0.01, ***p<0.001, ****p<0.0001.

The online version of this article includes the following source data and figure supplement(s) for figure 1:

**Source data 1.** Data for *Figure 1C and D*.

**Source data 2.** Data for *Figure 1E* and *Figure 1—figure supplement 2*.

**Source data 3.** Data for *Figure 1G*.

**Source data 4.** Data for *Figure 1H*.

*Figure 1 continued on next page*

offers no food except for negligible bacterial trails at the patch boundary, which only become significant food sources after about 5–6 hours of growth at room temperature. Thus, our experiments are limited to 6 hours to keep *C. elegans'* motivation to feed from the patch high. Basic prey refuge models often presume predators are highly successful at capturing prey, who in turn have low escape success, leading to a focus on the timing of prey's emergence from refuge when predators seem to leave the area (*Sih, 1992*). However, because adult *C. elegans* rarely die from a single bite and can escape most bites (*Wilecki et al., 2015*; *Quach and Chalasani, 2022*), their coexistence in the food patch with *P. pacificus* presents a sustained rather than immediate survival risk. Thus, our study will examine the use of both patch and refuge areas to establish the defense modes in our model system of nematode defensive behaviors.

In the circa-strike mode, we outline a three-step behavioral sequence: (1) escape a bite, (2) exit the patch, and (3) reenter the patch (*Figure 1A*). During the escape response, where *C. elegans* instinctively and rapidly accelerates away from a touch stimulus (*Pirri and Alkema, 2012*), *C. elegans* is unlikely to consider the patch and refuge in this first phase of the circa-strike. However, the subsequent phases involve deciding whether to move between the patch and refuge. Our previous findings show that *C. elegans* often exits the patch after being bitten by RS5194 *P. pacificus* (*Quach and Chalasani, 2022*), suggesting that the escape phase is often but not always followed by the exit phase of the circa-strike mode. In our experimental setup, we use an arena (*Figure 1*, *Figure 1—figure supplement 1A*) to confine *C. elegans* to a space with a bacterial patch as the only food source, necessitating its eventual reentry into the patch and thereby ensuring that the exit phase is always followed by the reentry phase. Importantly, the arena is wide relative to the small bacterial food patch placed in the center of the arena (*Figure 1*, *Figure 1—figure supplement 1A*), ensuring ample empty space around the patch for *C. elegans* to retreat to as refuge. In this arena, we placed one *C. elegans* and four RS5194 *P. pacificus*. To focus on innate behaviors, we observed behaviors for just 1 hour. Under these conditions, *C. elegans* exits the patch more often and spends significantly more time outside it in the presence of predators, in contrast to minimal exits and time spent outside the patch when predators are absent (*Figure 1C and D*). This indicates that *C. elegans* rarely leaves the patch unless provoked by a predatory attack.

In the post-encounter mode, we examine the feeding posture of *C. elegans* after extended exposure to a predator-inhabited patch (*Figure 1A*). In a previous study, we demonstrated that *C. elegans* tends to stay within the food patch for the first half-hour of exposure to a predator-inhabited patch (*Quach and Chalasani, 2022*). However, its behavior shifts over 6 hours, with *C. elegans* predominantly feeding with only its head in contact with the patch (*Quach and Chalasani, 2022*). In the current study, we define the outstretched feeding posture as *C. elegans* having its mouth contacting the patch boundary or a bacterial trail emanating from the patch while the rest of its body stretches outside of the patch (*Figure 1*, *Figure 1—figure supplement 1B*). This outstretched feeding posture allows for quick withdrawal from the patch in response to bites, while maintaining access to food and reducing the risk of predator detection. To evaluate post-encounter behavior, we use the same arena setup as in the circa-strike mode (*Figure 1B*), with hourly observations over six hours to monitor the prevalence of the outstretched feeding posture among *C. elegans*. To focus on feeding posture decisions, *C. elegans* was allowed time to settle into a stable feeding posture if it was transitioning between patch and refuge spaces. Our findings show an increased adoption of the outstretched posture in the

presence of predators (*Figure 1E*), which intensifies with prolonged predator exposure (*Figure 1—figure supplement 2*). This indicates that *C. elegans* learns to associate the patch with higher predation risk, opting to limit full entry into the patch as a defensive strategy.

In the pre-encounter mode, we studied how *C. elegans* approaches a new, predator-free after extended experience with a predator-inhabited patch (*Figure 1A*). We modified the light-dark transition test for unconditioned anxiety in rodents (*Crawley and Goodwin, 1980*; *Crawley, 1985*) to suit nematodes in our patch-refuge context. While the light-dark transition test measures exploration between a dark chamber and an aversive, brightly lit chamber, our adaptation measures exploration from an empty chamber into a chamber filled with a bacterial food patch. Unlike the light-dark transition test, where the brightly lit chamber inherently repels mice, the patch is not aversive to *C. elegans* unless it becomes associated with predation risk.

To determine if *C. elegans* takes into account its own vulnerability in addition to predation risk, we utilized two spatial configurations of patch-refuge: one that permits *C. elegans* to leave the patch from any point along its boundary (wide exit arena, same arena as for circa-strike and pre-encounter modes) and another that restricts exits to a narrow opening on the boundary (narrow exit arena) (*Figure 1F*, *Figure 1—figure supplement 1*). Critically, the narrow opening is small enough that it can be blocked by a predator, occasionally preventing *C. elegans* from exiting the patch. During the exposure period, we exposed *C. elegans* to predator-inhabited patches in either the wide exit or narrow exit arena, for either 2 or 4 hours (*Figure 1F*). As a mock control, *C. elegans* were exposed to these conditions, but without predators. Afterward, we tested pre-encounter behavior by transferring *C. elegans* to a new predator-free arena of the same type and measuring its latency to enter the new patch and the number of bins it explores on the patch within 15 minutes upon entry (*Figure 1F*, *Figure 1—figure supplement 1*). To ensure *C. elegans* has no prior awareness of predator presence in the new patch, we consistently placed it in the center of the empty chamber as its starting position. We hypothesized that previous experience with a predator-inhabited patch would lead *C. elegans* to approach and explore a new patch more cautiously, particularly when escape options are restricted. Our observations confirmed this, noting a significant delay in entering new patches exclusively after *C. elegans* spent 4 hours in a narrow exit arena with predators (*Figure 1G*). Moreover, we detected a decrease in exploration activity following just 2 hours of predator exposure in the narrow exit arena, with exploration diminishing further after 4 hours in both wide and narrow arenas (*Figure 1H*). Consequently, we decided to exclusively use the narrow exit arena in subsequent pre-encounter mode experiments. To explore whether delayed entry and diminished exploration of the patch resulted from mobility issues caused by predator-induced injuries, we measured the locomotor speed of predator-exposed and mock-exposed *C. elegans*. Given that *C. elegans* tends to move more quickly on bacteria-free surfaces, we reasoned that assessing speed before *C. elegans* enters the new patch would provide a clearer indication of any locomotion defects. Our findings revealed no noticeable difference in locomotor speed, indicating that exposure to predators did not affect *C. elegans*' mobility (*Figure 1—figure supplement 3A–C*). Furthermore, after spending 6 hours in a predator-free patch, the behavior of predator-exposed *C. elegans* returned to typical exploration patterns (*Figure 1—figure supplement 3D*), demonstrating intact mobility as well as ability to adjust behavior based on changes in experience. These results suggest that extended exposure to a predator-inhabited patch leads *C. elegans* to adopt a more cautious approach when exploring new, predator-free patches.

We omitted the narrow exit arena from our analysis of the circa-strike and post-encounter modes to avoid the possibility that restricted access could conceal *C. elegans*' efforts to seek refuge when predators are nearby. Our primary interest lies in discerning the prey's intent to use the patch or refuge, not in the obstacles imposed by particular patch-refuge layouts. However, because pre-encounter behaviors occur in the absence of predators, the use of a narrow exit arena did not interfere with our assessment of *C. elegans*' inclination toward patch or refuge use.

Given that both post-encounter and pre-encounter defense modes involve significant time with predator-occupied patches, the observed behaviors may be due to food scarcity from avoiding the patch, rather than actual defensive responses. To explore this possibility, we subjected food-deprived *C. elegans* to post-encounter and pre-encounter conditions without predators. Prey refuge theory predicts that food deprivation should lead to increased patch use and reduced refuge use, the opposite of what predator presence would cause (*Sih, 1992*). In post-encounter scenarios, food-deprived, non-predator-exposed *C. elegans* rarely adopted the outstretched feeding posture, unlike well-fed,

predator-exposed counterparts who frequently did after 2 hours (*Figure 1—figure supplement 4A*). For pre-encounter conditions, food-deprived, non-predator-exposed *C. elegans* entered new patches faster than well-fed, predator-exposed animals (*Figure 1—figure supplement 4B*), aligning with predictions that food deprivation increases patch use and decreases refuge use. However, the amount of patch explored by food-deprived, non-predator-exposed *C. elegans* was similar to that of well-fed, predator-exposed animals (*Figure 1—figure supplement 4C*). Considering that food-deprived *C. elegans* more dramatically slows down upon finding food compared to well-fed animals (*Sawin et al., 2000*), the interpretation of patch exploration is complicated in the absence of other evidence. However, these results as a collective suggest that food deprivation alone does not explain the defensive behaviors in our model's post-encounter and pre-encounter modes. This conclusion is consistent with our previous finding that *C. elegans* remains feeding, with its mouth in contact with the bacteria, throughout extended periods in predator-occupied patches, regardless of whether the rest of its body is inside the patch (*Quach and Chalasani, 2022*).

## Defensive response intensity increases with predation risk

We next investigated the sensitivity of nematode defensive modes to different levels of predation risk, which allowed us to further refine our behavioral metrics. Prey refuge theory suggests that prey will increasingly avoid areas where predators pose a greater danger, leading to reduced patch use and increased refuge use (*Sih, 1992*). To confirm this in our model system, we tested four strains of *Pristionchus* spp. nematodes, each representing a qualitatively different level of threat to *C. elegans*: TU445 (non-aversive bite), JU1051 (aversive but nonlethal bite), PS312 (aversive, potentially lethal within 24 hours), and RS5194 (aversive, potentially lethal within 4 hours) (*Figure 2A*). The TU445 strain, a *P. pacificus eud-1* mutant, exhibits a non-predatory mouthform whose bites are largely non-aversive to adult *C. elegans* (*Ragsdale et al., 2013*; *Wilecki et al., 2015*). JU1051, on the other hand, can deliver aversive bites but cannot kill adult *C. elegans* (*Pribadi et al., 2023*). PS312, the standard *P. pacificus* strain, poses a 50% chance of killing adult *C. elegans* within 24 hours in a bacteria-free, refuge-free environment (*Quach and Chalasani, 2022*). RS5194 *P. pacificus*, more lethal, has a similar fatality rate within just 4 hours in the same bacteria-free, refuge-free environment, increasing to around 70% by 8 hours (*Quach and Chalasani, 2022*). Considering our experiments involve up to 6 hours of predator exposure, only RS5194 poses a significant, timely threat to *C. elegans* survival in our model. To minimize harm and survive long-term exposure to RS5194 *P. pacificus*, *C. elegans* must adopt defensive strategies and utilize refuges effectively.

We first investigated how various predators influence circa-strike behavior. To confirm that the aversive nature of bites, rather than merely the presence of predators, triggers *C. elegans* to escape and exit the patch, we counted the instances of both spontaneous and bite-induced escapes and exits. Indeed, encounters with the non-aversive TU445 resulted in very low numbers of escapes and exits compared to strains with aversive bites (, *Figure 2B*, *Figure 2—figure supplement 1A*). Consequently, TU445 was excluded from further analysis of circa-strike behaviors that are conditional on escape and exit events. Following bite-induced escapes, *C. elegans* showed similar exit latencies across all aversive predator strains (*Figure 2—figure supplement 1B*). Nonetheless, the critical factor appears to be the decision to exit rather than the speed of doing so. Our findings indicated a higher likelihood of *C. elegans* exiting the patch after being bitten by RS5194 compared to JU1051 or PS312 (*Figure 2C*). Furthermore, *C. elegans* sometimes aborts exiting the patch after protruding its head outside of the patch, suggesting that the patch-to-refuge transition is a critical decision point (*Figure 2—figure supplement 1C*). We also evaluated the time it took for *C. elegans* to reenter to the patch after exiting. Similar to exit latency, we found that reentry latencies were consistent across all aversive predator strains (*Figure 2D*). These observations reveal that the decision to leave the patch after an escape response is a more precise indicator of predation risk's impact on circa-strike behavior than the metrics of how quickly exits or reentries occur.

Next, we assessed the impacts of various predators on post-encounter behavior. We first checked whether post-encounter behavior is specifically triggered by aversive bites. As expected, only a small percentage of *C. elegans* animals adopted the outstretched feeding posture in response to non-aversive TU445 predators, similar to that observed in the absence of predators (*Figure 2E*). When exposed to JU1051 and PS312, *C. elegans* demonstrated an intermediate prevalence of outstretched feeding, with both predators eliciting similar responses across the duration of predator exposure

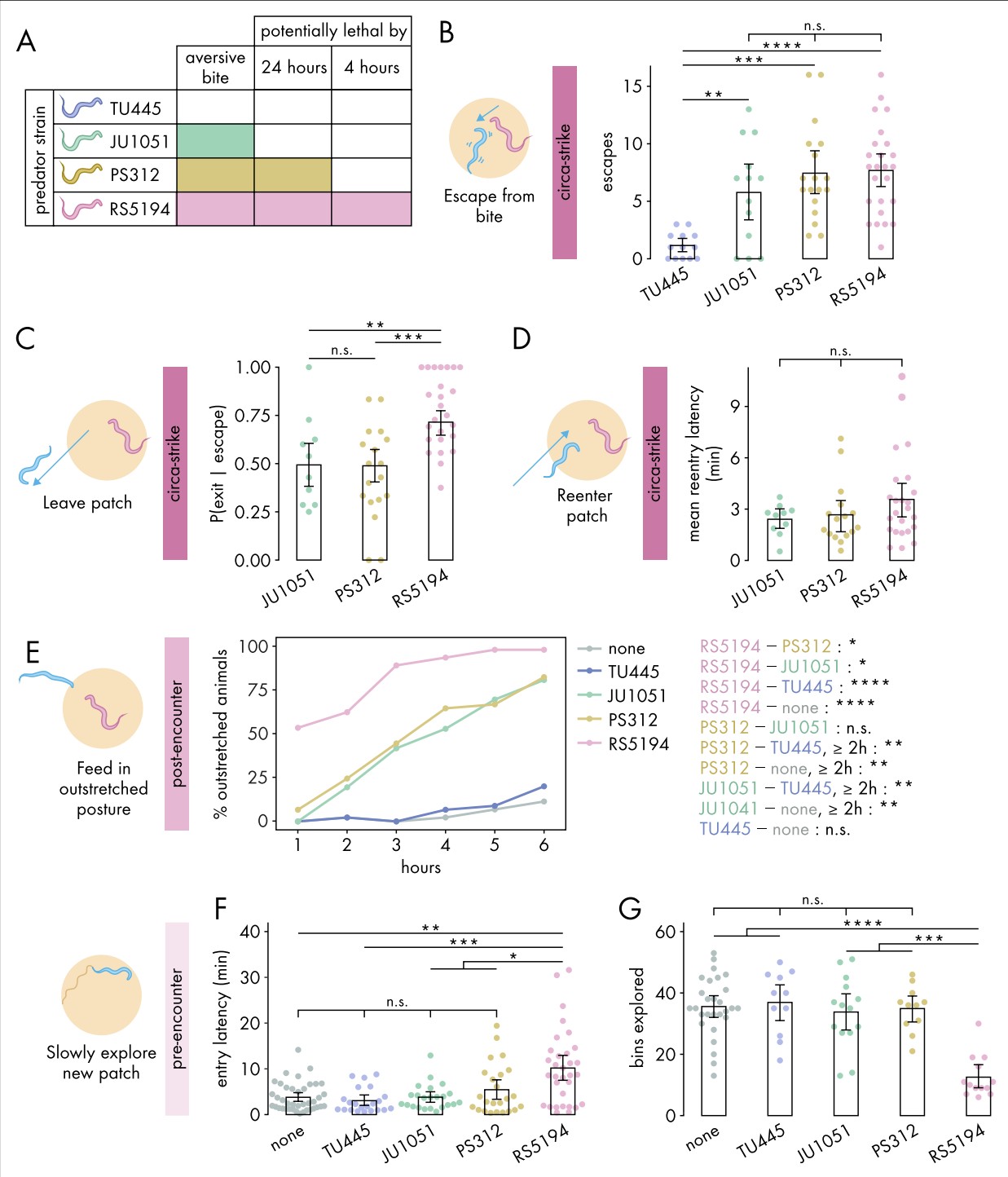

**Figure 2.** Defensive response intensity increases with predation risk. (**A**) Schematic of predatory harm potential of various *Pristionchus* spp. predator strains, based on studies previously conducted by this lab. (**B**) Number of bite-induced escape responses during 1-hour exposure to *Pristionchus* spp. predators (Dunn's test with Benjamini–Hochberg adjustment, n$_{C.\ elegans}$ = 13–25). (**C**) Probability of exit following an escape response (binomial logistic regression followed by Wald test with single-step adjustment for Tukey contrasts, n$_{C.\ elegans}$ = 10–25). (**D**) Latency to reenter the patch following an exit, averaged across escape-induced exits, for various predator strains (Kruskal–Wallis test, n$_{C.\ elegans}$ = 10–25). (**E**) Percentage of *C. elegans* animals adopting outstretched feeding posture across different exposure durations to various predator conditions (Fisher's exact test with Benjamini–Hochberg adjustment, n$_{C.\ elegans}$ = 36–45). Pairwise comparisons between predator strains are displayed on the right. (**F**) Latency to enter a new patch (Dunn's test with Benjamini–Hochberg adjustment, n$_{c.\ elegans}$ = 22–40) and (**G**) bins explored (Dunn's test with Benjamini-Hochberg adjustment, n$_{C.\ elegans}$ = 11–28)

*Figure 2 continued on next page*

*Figure 2 continued*
following 4-hour exposure to various predator conditions. Error bars in (**C**) are predicted P(exit | escape) and 95% CIs from binomial logistic regression model of data. All other error bars are 95% bootstrap CIs containing the mean. n.s. = p>0.05, *p<0.05, **p<0.01, ***p<0.001, ****p<0.0001.

The online version of this article includes the following source data and figure supplement(s) for figure 2:

**Source data 1.** Data for *Figure 2B–D*, *Figure 2—figure supplement 1B and C*.

**Source data 2.** Data for *Figure 2E*.

**Source data 3.** Data for *Figure 2F*.

**Source data 4.** Data for *Figure 2G*, *Figure 2—figure supplement 2A and B*.

**Figure supplement 1.** Patch exit latency is unaffected by predator strain.

**Figure supplement 1—source data 1.** Data for *Figure 2—figure supplement 1A*.

**Figure supplement 2.** Effect of extended exposure to various predations on pre-encounter behavior.

(*Figure 2E*). In comparison, RS5194 triggered the most pronounced increase in outstretched feeding, distinguishing itself by eliciting a significant increase in outstretched feeding as early as the 1-hour time point (*Figure 2E*). Despite PS312's potential lethality within 24 hours and JU1051's non-lethal nature, their elicited post-encounter responses were similar, indicating *C. elegans* perceives them as comparable threats over the 6-hour exposure period. Conversely, RS5194's potential lethality within this timeframe likely accounts for the heightened post-encounter behavioral adjustments. Thus, these findings indicate a tiered post-encounter response based on predation risk: non-aversive, aversive but not imminently lethal, and potentially lethal within the exposure period.

Lastly, we evaluated how various predators affect pre-encounter behavior. We utilized the narrow exit arena for both the exposure and testing periods to effectively induce pre-encounter responses. Similar to what we observed in the circa-strike and post-encounter scenarios, interactions with non-aversive TU445 predators yielded responses akin to those in predator-free conditions, including similar delays in entering a new patch and exploration levels within it (*Figure 2F and G*). However, unlike in circa-strike and post-encounter behaviors, exposure to JU1051 and PS312 predators also resembled exposure to non-aversive predators and predator-free conditions (*Figure 2—figure supplement 2*). The exception was RS5194, which uniquely caused *C. elegans* to delay entering the patch and to reduce exploration upon entry following a 4-hour exposure period (*Figure 2D*). With extended exposure, *C. elegans* facing PS312—but not JU1051 or TU445—also exhibited less exploration compared to predator-free conditions (*Figure 2—figure supplement 2*). These findings suggests that pre-encounter behavior primarily emerges in response to predators posing a direct threat to life, with the behavior developing more rapidly under threat from more lethal predators. Overall, our results show that RS5194 *P. pacificus*, which represents a significant lethal risk within the duration of our behavioral experiments, consistently elicits the strongest responses across circa-strike, post-encounter, and pre-encounter modes. Based on its ability to elicit all three defense modes, we selected RS5194 as the predator strain for use in subsequent experiments.

## SEB-3 and NLP-49 peptides differentially regulate defense modes

After we confirmed that our nematode defensive behavior model can effectively detect responses to different levels of predation risk, we examined whether the different defense modes are associated with distinct underlying molecular mechanisms. In *C. elegans*, the *seb-3* gene encodes the SEB-3 receptor, while the *nlp-49* gene encodes two peptides, NLP-49-1 and NLP-49-2. Currently, the only known ligand for SEB-3 is NLP-49-2, while no receptor is currently known to be activated by NLP-49-1 (*Chew et al., 2018*; *Beets et al., 2023*). Previous studies have shown that changes in *seb-3* and *nlp-49* expression result in coordinated changes in various behaviors, suggesting that SEB-3 directly interacts with NLP-49-2 to influence these behaviors (*Chew et al., 2018*). To see if this is also the case in our model system of nematode defensive behaviors, we tested deletion mutants with the alleles *seb-3(tm1848)* and *nlp-49(gk546875)*, as well as *seb-3* and *nlp-49* overexpression strains. The *seb-3* and *nlp-49* overexpression strains are transgenic lines that were generated by microinjection (*Mello et al., 1991*), resulting in extrachromosomal arrays containing many copies of *seb-3* or *nlp-49*, whose expression is driven by their endogenous promoters. A previous study has shown that *seb-3* and

*nlp-49* overexpression strains generated in this manner exhibit phenotypes that are opposite of *seb-3* and *nlp-49* deletion mutants (*Chew et al., 2018*).

Before assessing the defense modes of *seb-3* and *nlp-49* strains, we first checked for changes in baseline locomotor speeds that may affect interpretation of circa-strike behaviors. We first measured the baseline speed on bacterial surfaces following a bite, critical for understanding exit latencies, using an arena that blocks *C. elegans* from exiting the patch (*Figure 3—figure supplement 1A*). Based on when most exits occur following to a bite (*Figure 3—figure supplement 1B and C*), we measured the average on-bacteria escape speed for 15 seconds post-bite. Since escape responses can habituate with repeated stimulation (*Rankin et al., 1990*), we examined on-bacteria escape speed across consecutive bites. We found that escape speeds for *seb-3* strains matched those of wildtype animals (*Figure 3—figure supplement 1D*), consistent with past findings that *seb-3* loss or gain-of-function mutations do not significantly alter speed after mechanical stimulation (*Jee et al., 2013*). However, *nlp-49* overexpression animals displayed sustained escape speeds across bites, compared to wildtype, indicating slower habituation (*Figure 3—figure supplement 1E*). This is consistent with previous findings that *nlp-49* overexpression animals have increased baseline speed during spontaneous locomotion on bacteria (*Chew et al., 2018*). Next, we examined baseline speed of *C. elegans* in a wide arena devoid of bacteria and predators, relevant for interpreting reentry latencies. Based on when most reentries occur following an exit (*Figure 2D*), we measured the average speed across 5 minutes of exploration. In line with prior findings (*Jee et al., 2013*), *seb-3* deletion mutants showed no significant speed difference from wildtype in these conditions (*Figure 3—figure supplement 1F*). Similarly, *nlp-49* deletion mutants and overexpression animals, exhibited speeds comparable to wildtype (*Figure 3—figure supplement 1F and G*). However, *seb-3* overexpression animals moved slower than wildtype (*Figure 3—figure supplement 1G*). Thus, the heightened baseline speed of *nlp-49* overexpression animals may affect interpretation of exit latencies, while the reduced baseline speed of *seb-3* overexpression animals may affect interpretation of reentry latencies.

Taking this into consideration, we examined the roles of *seb-3* and *nlp-49* in regulating the circa-strike defense mode. We found that *seb-3* and *nlp-49* strains executed similar numbers of bite-induced escape responses as wildtype (*Figure 3—figure supplement 2A and B*), suggesting that these strains have similar sensitivity to bites as wildtype animals. All *seb-3* and *nlp-49* strains displayed exit latencies similar to wildtype animals (*Figure 3—figure supplement 2C and D*), despite the increased bite escape speed phenotype of *nlp-49* overexpression animals (*Figure 3—figure supplement 1E*). We found that *seb-3* overexpression animals were less likely than wildtype animals to exit a patch following a bite-induced escape response (*Figure 3A*), but did not see this effect mirrored in *nlp-49* overexpression animals (*Figure 3B*). Unlike the divergent effects of *seb-3* and *nlp-49* on exit probability, we found that changes in *nlp-49* and *seb-3* expression resulted in similar changes to reentry latencies (*Figure 3C and D*). Both *seb-3* and *nlp-49* deletion mutants displayed longer reentry latencies compared to wildtype, while both *seb-3* and *nlp-49* overexpression strains exhibited shorter reentry latencies (*Figure 3C and D*). The longer reentry latencies of *seb-3* and *nlp-49* deletions mutants are not explained by slower baseline speeds as both have similar baseline speeds as wildtypes animals in bacteria-free, predator-free environments (*Figure 3—figure supplement 1F and G*). Similarly, the shorter reentry latency phenotypes in *seb-3* and *nlp-49* overexpressions strains are not explained by faster baseline speeds as these strains show either similar or slower speeds compared to wildtype (*Figure 3—figure supplement 1F and G*). Thus, the reentry phenotypes observed in *seb-3* and *nlp-49* deletion mutants seem to be predator-induced and suggest increased defensive response compared to wildtype animals. Overall, these results suggest that within the circa-strike mode, the exit phase is predominantly regulated by *seb-3*, while the reentry phase is regulated by both *seb-3* and *nlp-49* to similar effects.

Next, we explored how *seb-3* and *nlp-49* function in the post-encounter defense mode. While *seb-3* deletion mutants were similar to wildtype, *nlp-49* deletion mutations exhibited increased outstretched feeding in the first 2 hours (*Figure 3E and F*). To see if a ceiling effect occluded differences between *nlp-49* deletion mutants and wildtype at later hours, we also evaluated outstretched feeding posture on a higher density bacterial patch. Since RS5194 *P. pacificus* predators bite less on higher density bacterial patches (*Quach and Chalasani, 2022*), we reasoned that prevalence of outstretched feeding observed in wildtype *C. elegans* would decrease accordingly, providing a clear comparison to observe differences between wildtype and *nlp-49* deletion mutants at later hours.

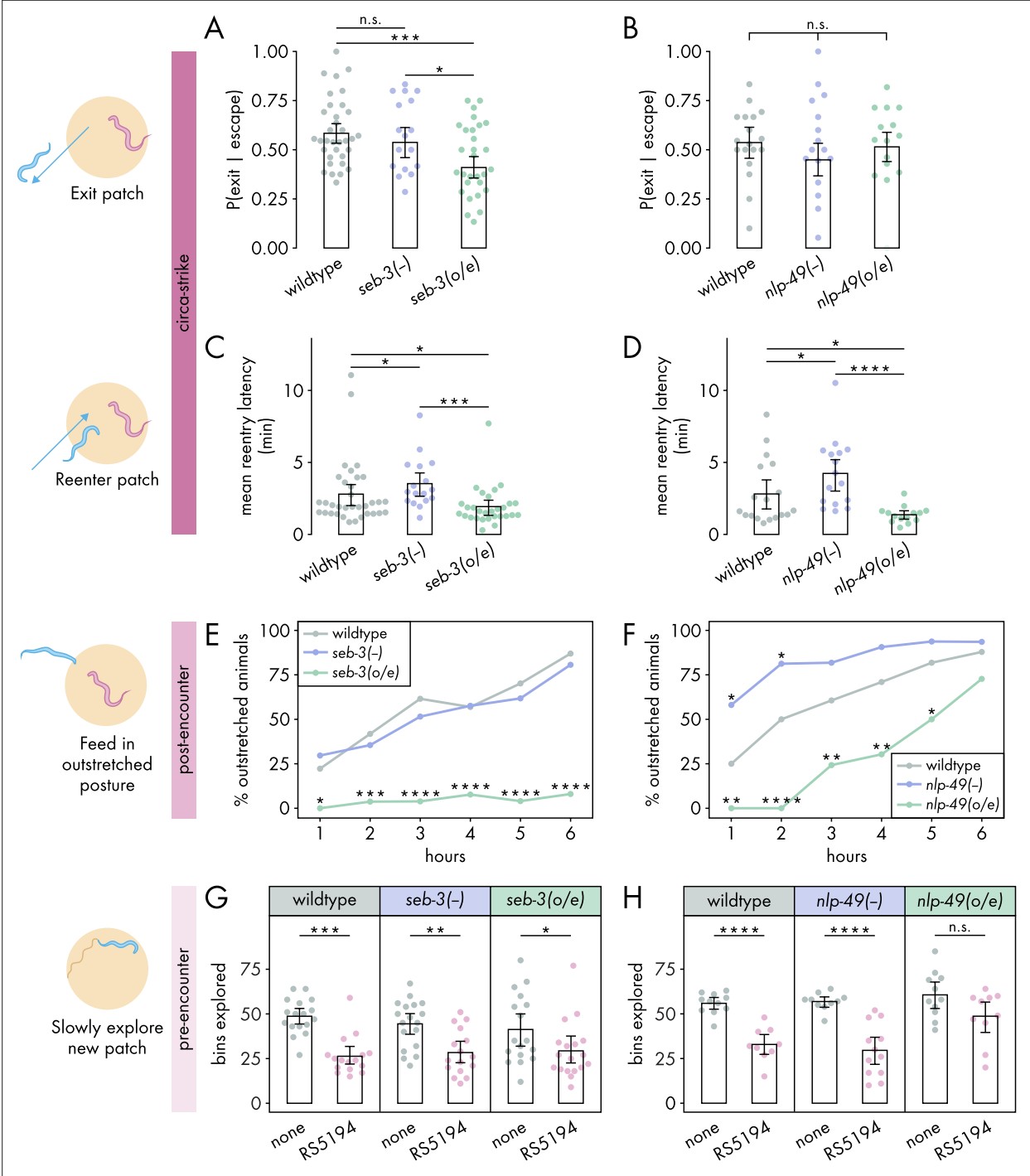

**Figure 3.** SEB-3 and NLP-49 peptides differentially regulate defense modes. (**A, B**) Probability of exit following an escape response by (**A**) *seb-3* strains (n$_{C. elegans}$ = 17–34) and (**B**) *nlp-49* strains (n$_{C. elegans}$ = 15–18) (binomial logistic regression followed by Wald test with single-step adjustment for Tukey contrasts). (**C, D**) Latency to reenter the patch following an exit, averaged across escape-induced exits, for (**C**) *seb-3* strains (n$_{C. elegans}$ = 17–34) and (**D**) *nlp-49* strains (n$_{C. elegans}$ = 14–18) (Dunn's test with Benjamini–Hochberg adjustment). (**E, F**) Percentage of animals adopting outstretched feeding posture in (**E**) *seb-3* strains (n$_{C. elegans}$ = 27–63) and (**F**) *nlp-49* strains (n$_{C. elegans}$ = 31–33) (Fisher's exact test with Benjamini–Hochberg adjustment). (**G, H**) Bins explored following 4-hour exposure to predator or predator-free conditions, by (**G**) *seb-3* strains (Wilcoxon's ranked sum test with Benjamini–Hochberg adjustment, n$_{C. elegans}$ = 16–18) and (**H**) *nlp-49* strains (Welch's *t*-test with Benjamini–Hochberg adjustment, n$_{C. elegans}$ = 9–12). Error bars in (**A, B**) are predicted P(exit | escape) and 95% CIs from binomial logistic regression model of data. All other error bars are 95% bootstrap CIs containing the mean. n.s. = p > 0.05, *p<0.05, **p<0.01, ***p<0.001, ****p<0.0001.

The online version of this article includes the following source data and figure supplement(s) for figure 3:

*Figure 3 continued on next page*

*Figure 3 continued*

**Source data 1.** Data for *Figure 3A and C*, *Figure 3—figure supplements 1B and 2A, C*.

**Source data 2.** Data for *Figure 3B and D*, *Figure 3—figure supplements 1C and 2B, D*.

**Source data 3.** Data for *Figure 3E*.

**Source data 4.** Data for *Figure 3F*.

**Source data 5.** Data for *Figure 3H*.

**Source data 6.** Data for *Figure 3H*.

**Figure supplement 1.** Baseline speeds of *seb-3* and *nlp-49* strains.

**Figure supplement 1—source data 1.** Data for *Figure 3—figure supplement 1D*.

**Figure supplement 1—source data 2.** Data for *Figure 3—figure supplement 1E*.

**Figure supplement 1—source data 3.** Data for *Figure 3—figure supplement 1F, G*, *Figure 4—figure supplement 2A*.

**Figure supplement 2.** Escapes and exit latencies by *seb-3* and *nlp-49* strains.

**Figure supplement 3.** Outstretched feeding on high-density bacteria by *nlp-49* strains.

**Figure supplement 3—source data 1.** Data for *Figure 3—figure supplement 3*.

**Figure supplement 4.** New patch entry latency by *seb-3* and *nlp-49* strains.

**Figure supplement 4—source data 1.** Data for *Figure 3—figure supplement 4A and C*.

**Figure supplement 4—source data 2.** Data for *Figure 3—figure supplement 4B and D*.

---

Indeed, our observations revealed that in higher density patches, wildtype animals did not display saturating levels of outstretched feeding behavior as they did on lower density bacterial patches (*Figure 3F*, *Figure 3—figure supplement 3*). Relative to this reduced wildtype behavior, *nlp-49* deletion mutants exhibited significantly higher prevalence of animals in outstretched feeding posture at all timepoints (*Figure 3—figure supplement 3*). While *seb-3* deletion mutants behave similarly to wildtype, *seb-3* overexpression animals consistently exhibited a near-zero prevalence of outstretched feeding posture throughout the 6-hour exposure period (*Figure 3E*). Similar to *seb-3* overexpression animals, *nlp-49* overexpression animals also showed reduced outstretched feeding compared to wildtype (*Figure 3F*). Altogether, these results suggest that, unlike *seb-3* deletion, *nlp-49* deletion has a positive modulatory effect on post-encounter behavior. In contrast, overexpression of *seb-3* and *nlp-49* both suppress outstretched feeding.

Finally, we investigated the roles of *seb-3* and *nlp-49* strains in the pre-encounter defense mode. To account for potential influences of strain-specific differences in baseline locomotion, we compared predator-exposed and mock-exposed (same setup without predators) animals within each strain. We first evaluated the latency to enter a new predator-free patch following a 4-hour predator exposure period in a narrow exit arena. In all within-strain comparisons, predator-exposed animals delayed entering the new patch longer than mock-exposed animals (*Figure 3—figure supplement 4A and B*), suggesting that changes in *seb-3* or *nlp-49* expression alone were not sufficient to suppress the entry latency of predator-exposed animals to that of mock-exposed levels. Thus, we next compared across strains to look for more subtle effects. To justify comparing predator-exposed animals across strains, we first checked that mock-exposed animals were comparable across strains. Under mock conditions, *nlp-49* deletion mutants significantly differed wildtype animals (*Figure 3—figure supplement 4C and D*), so we excluded *nlp-49* deletion mutants from our analysis of predator-exposed animals. Comparing only predator-exposed animals, we found that *seb-3* deletion mutants and overexpression strains both exhibited entry latencies similar to that of wildtype animals (*Figure 3—figure supplement 4C*). In contrast, predator-exposed *nlp-49* overexpression animals entered the new patch sooner than predator-exposed wildtype animals (*Figure 3—figure supplement 4D*). However, the gathering of most entry latency values near zero for mock-exposed *nlp-49* overexpression and wildtype animals suggests the possibility of a floor effect, so we may not be able to observe a sub-wildtype mock phenotype in our setup if one exists for *nlp-49* overexpression animals. We next looked at the number of bins explored by *C. elegans* once it entered the new patch. This metric has a more dynamic range for both mock- and predator-exposed wildtype animals (*Figure 1H*), so interpretations of effects should be more robust. Predator-exposed *nlp-49* overexpression animals explored the new patch similarly to mock-exposed animals, while the predator-exposed animals of wildtype and all

other *seb-3* and *nlp-49* strains explored the patch less than corresponding mock-exposed animals (*Figure 3G and H*). This patch exploration phenotype of *nlp-49* overexpression animals is consistent with its shorter latency phenotype, both suggesting that NLP-49 peptides suppress pre-encounter behavior. Meanwhile, SEB-3 seems to have no direct effect on pre-encounter behavior.

Taking into account all defense modes, SEB-3 and NLP-49 peptides are each involved in regulating at least two of the three defense modes. However, the divergent effects of SEB-3 and NLP-49 peptides suggest that their regulation of defensive behaviors involves signaling interactions other than NLP-49-2 directly binding the SEB-3 receptor. Furthermore, NLP-49 peptides appear not to be directly involved in an early phase of the defense phase associated with the highest predatory imminence (circa-strike, exit phase), while SEB-3 seems to have no direct role in the defense mode with the least predatory imminence (pre-encounter).

## Interdependence between SEB-3 and NLP-49 peptides shifts across defense modes

To further investigate interdependence between SEB-3 and NLP-49 peptides in regulating defensive behaviors, we tested mutants with double deletions in *seb-3* and *nlp-49*, as well as a *nlp-49* overexpression animals lacking *seb-3*. These strains have been previously used to assess interdependence between SEB-3 and NLP-49 peptides in regulating other types of stress behaviors (*Chew et al., 2018*).

We first explored the interdependence of SEB-3 and NLP-49 peptides in regulating the circa-strike mode. In the exit phase of the circa-strike mode, double deletion mutants were similar to wildtype in number of escapes and exit latency (*Figure 4—figure supplement 1A and B*). While *nlp-49* deletion and *seb-3* deletion mutants individually did not affect the probability of *C. elegans* exiting a patch following a bite-induced escape (*Figure 3A and B*), we found that double deletion mutants exhibited decreased exit probability compared to wildtype (*Figure 4A*). This decrease was abolished in *nlp-49* overexpression animals lacing *seb-3* (*Figure 4A*). These results suggest that SEB-3 and NLP-49 peptides may independently contribute to the regulation of the exit phase of the circa-strike mode, with the possibility of a compensatory or synergistic interaction. Next, we assessed the reentry phase of the circa-strike mode. If the similar increased reentry latency phenotypes of *seb-3* and *nlp-49* single deletion mutants are due to direct interaction between SEB-3 and NLP-49–2, then we would expect double deletion mutants to have a similar reentry latency as the *seb-3* single deletion mutant since the removal of one binding partner should be sufficient to preclude the function of both. Similarly, we would also expect *seb-3* single deletion mutants with *nlp-49* overexpression to have a similar reentry latency as the *seb-3* single deletion mutant since additional NLP-49-2 should have no effect without its binding partner SEB-3. However, we found that double deletion mutants and *nlp-49* overexpression lacking *seb-3* exhibited reentry latencies similar to that of wildtype and lower than that of *seb-3* single deletion mutants (*Figure 4B*). These differences in reentry latencies are not due to altered locomotor speed as all strains have similar baseline speeds in off-bacteria, predator-free conditions (*Figure 4—figure supplement 2A*). This suggests that the reentry latency phenotype of *nlp-49* or *seb-3* single deletion mutants are dependent on the normal expression of the other gene. Additionally, disruption of normal expression of both genes in the same animal results in wildtype reentry latency. Overall, these results indicate that SEB-3 and NLP-49 peptides likely interact in a complex manner to regulate circa-strike behavior, with both genes playing distinct but interconnected roles in this process.

We next investigated the interdependence of SEB-3 and NLP-49 peptides in the post-encounter mode. We first explored whether the enhanced outstretched feeding phenotype of *nlp-49* deletion mutants and the wildtype phenotype in *seb-3* deletion mutants (*Figure 3E and F*) indicate that SEB-3 and NLP-49 peptides act independently of each other. If so, then double deletion mutants should have a similar phenotype to *nlp-49* deletion mutants. However, we observed that the enhanced outstretched feeding phenotype of *nlp-49* single deletion mutants was abolished in double mutants, which instead more closely resembled *seb-3* single deletion mutants (*Figure 4C*). This suggests that the phenotype observed in the *nlp-49* single deletion mutants is not solely mediated by NLP-49 peptides acting independently of SEB-3. Next we looked at whether the reduced outstretched feeding phenotype of *nlp-49* overexpression animals depends on SEB-3 (*Figure 3E*). If this phenotype is entirely dependent on SEB-3, then we would expect the phenotype to be fully abolished in *seb-3* single deletion mutants with *nlp-49* overexpression. Instead, this strain exhibited outstretched feeding levels that are intermediate between those of *seb-3* deletion or *nlp-49* overexpression alone (*Figure 4D*). Overall, these

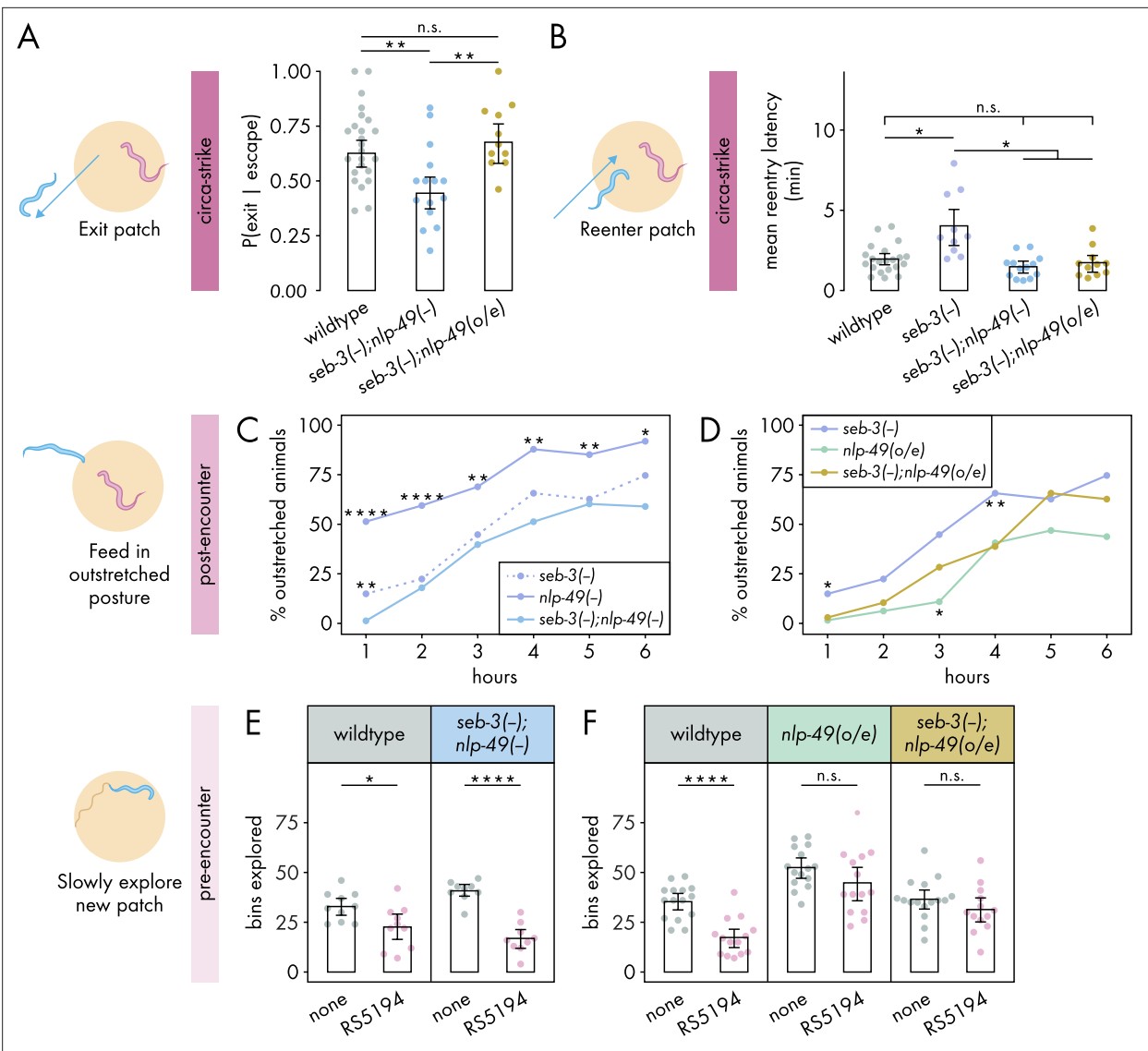

**Figure 4.** Interdependence between SEB-3 and NLP-49 peptides shifts across defense modes. (**A**) Probability of exit following an escape response by *seb-3* deletion strains with *nlp-49* deletion or overexpression (binomial logistic regression followed by Wald test with single-step adjustment for Tukey contrasts, n$_{C. elegans}$ = 11–24). (**B**) Latency to reenter the patch following an exit, averaged across escape-induced exits, for strains with *seb-3* deletion by itself or with *nlp-49* deletion or overexpression (Games–Howell test, n$_{C. elegans}$ = 11–21). (**C**) Percentage of animals adopting outstretched feeding posture in *seb-3* deletion strains with *nlp-49* deletion or overexpression (Fisher's exact test with Benjamini–Hochberg adjustment, n$_{C. elegans}$ = 67–78). Significance asterisks represent comparisons with *seb-3(-);nlp-49(-)* (blue). (**D**) Percentage of animals adopting outstretched feeding posture in *seb-3* deletion strains with *nlp-49* deletion or overexpression (Fisher's exact test with Benjamini–Hochberg adjustment, n$_{C. elegans}$ = 64–67). Significance asterisks represent comparisons with *seb-3(-);nlp-49(o/e)* (yellow). (**E**) Bins explored following 4-hour exposure to predator or predator-free conditions, by *seb-3* deletion strains with *nlp-49* deletion or overexpression (Student's *t*-test with Benjamini-Hochberg adjustment, n$_{C. elegans}$ = 9–10). (**F**) Bins explored following 4-hour exposure to predator or predator-free conditions, by *seb-3* deletions strains with or without *nlp-49* overexpression (Student's *t*-test with Benjamini–Hochberg adjustment, n$_{C. elegans}$ = 13–16). Error bars in (**A**) are predicted P(exit | escape) and 95% CIs from binomial logistic regression model of data. All other error bars are 95% bootstrap CIs containing the mean. n.s. = p>0.05, *p<0.05, **p<0.01, ***p<0.001, ****p<0.0001.

The online version of this article includes the following source data and figure supplement(s) for figure 4:

**Source data 1.** Data for *Figure 4A*, *Figure 4—figure supplement 1A and B*.

**Source data 2.** Data for *Figure 4B*.

**Source data 3.** Data for *Figure 4C*.

**Source data 4.** Data for *Figure 4D*.

**Source data 5.** Data for *Figure 4E*.

*Figure 4 continued on next page*

*Figure 4 continued*

**Source data 6.** Data for *Figure 4F*.

**Figure supplement 1.** Circa-strike behavior of *seb-3* deletion strains with *nlp-49* deletion or over expression.

**Figure supplement 2.** Baseline locomotion of *seb-3* deletion strains with *nlp-49* deletion or overexpression.

**Figure supplement 2—source data 1.** Data for *Figure 4—figure supplement 2B*.

findings suggest that although SEB-3 and NLP-49 peptides have distinct functions in influencing post-encounter behavior, they also exhibit a degree of interdependence in this modulation.

Finally, we assessed the interdependence of SEB-3 and NLP-49 peptides in the pre-encounter mode. While both *seb-3* and *nlp-49* single deletion mutants resemble wildtype, we wondered if double deletion mutants would differ from wildtype, as was observed for the exit phase of circa-strike mode (*Figure 4A*). However, predator-exposed double mutants explored a new patch less than mock-exposed double mutants, a pattern similar to wildtype animals (*Figure 4E*). This indicates a lack of compensatory interaction between SEB-3 and NLP-49 peptides, and that neither SEB-3 nor NLP-49 peptides are required to maintain wildtype pre-encounter response. Entry latency was excluded from our analysis due to the subtle effects of *nlp-49* overexpression, which required comparisons across strains. Such comparisons necessitated consistent behavior among mock-exposed animals from all strains, a criterion not satisfied by double mutants relative to wildtype (*Figure 4—figure supplement 2*). Consequently, we concentrated on the number of bins explored on a new patch as a more reliable measure of pre-encounter behavior. Next, we determined whether the reduced pre-encounter pheno-type of *nlp-49* overexpression animals (*Figure 3H*) is dependent on SEB-3. Similar to animals with only *nlp-49* overexpression, *nlp-49* overexpression animals lacking *seb-3* show no difference in their explo-ration of the new patch across predator-exposed and mock-exposed conditions (*Figure 4*). These results indicate that NLP-49 overexpression alone is sufficient to disrupt the typical pre-encounter response, and this effect is not influenced by the presence or absence of SEB-3. Thus, in the pre-encounter mode, NLP-49 peptides appears to act independently of SEB-3, suggesting distinct regu-latory mechanisms for the pre-encounter mode compared to other defense modes.

## Discussion

Our model system offers a comprehensive view of nematode defensive behaviors, focusing on the adaptive strategies *C. elegans* employs in utilizing patch and refuge spaces while foraging under predatory threat. This approach integrates principles from both prey refuge theory and predatory imminence theory. Predatory imminence theory helps us categorize defensive behaviors into distinct defense modes based on the spatiotemporal proximity of predatory attack, which dictates the urgency with which prey need to deal with predatory threat. Prey refuge theory, on the other hand, provides a framework for understanding the decision-making process of *C. elegans* as it navigates the trade-offs between feeding and safety when predation risk varies across space. This creates a consistent and predictable environment for examining how *C. elegans* should flexibly adjust its behavior in each defense mode to achieve optimal patch and refuge use. Our model delineates three defense modes that describe how *C. elegans* interacts with a predator-associated patch and a predator-free refuge, each representing different levels of predatory imminence, uncertainty, and experience (*Figure 1A*). In the circa-strike mode, *C. elegans* responds to a predatory bite by executing an escape response, exiting the patch, and then reentering the patch. With repeated experiences of bites and circa-strike responses, *C. elegans* learns to associate the patch with predation risk and develops anticipatory behaviors for future encounters in post- and pre-encounter defense modes. In the post-encounter mode, *C. elegans* assumes an outstretched feeding posture for feeding at the periphery of a patch that it knows to be inhabited by predators. In the pre-encounter defense mode, when approaching a new patch without specific knowledge of its safety, *C. elegans* is less quick to enter and explore the new patch, influenced by its accumulated experiences of predation risk associated with similar patches. These defense modes provide a narrative on how *C. elegans* might acquire and apply them in its natural life, shaped by experience and perceived predation risk. The consistent patch and refuge foraging environment across these modes allows for behavioral changes to be attributed to *C. elegans*'s changing perceptions and experiences, rather than external setup variations. This model

thus bridges the behaviors observed in *C. elegans* with underlying theories of predator–prey interactions, offering insights into the complex decision-making processes that nematode prey may face in the wild.

Our research demonstrates that only life-threatening predators trigger all three defense modes in nematode behavioral models, highlighting how *C. elegans* differentiates between non-threatening, aversive but nonlethal, and life-endangering threats through a variety of responses. Consistent with the function of escape responses as innate reflexes for immediate evasion of mechanosensory stimuli (*Pirri and Alkema, 2012*), *C. elegans* executes similar numbers of escape responses for all aversive predators in our study (*Figure 2B*). In contrast, our study shows that more nuanced behaviors are possible when *C. elegans* has more time to make a behavioral choice, especially in decision-making contexts in with food access and predation risk are conflicting factors. In the circa-strike mode (exit phase) and the post-encounter mode, responses are split into three tiers of intensity: minimal response to non-aversive predators, intermediate responses to aversive but nonlethal predators (including predators that are lethal on a irrelevantly long timescale), and maximal responses to predators that are life-threatening within the timescale of the behavioral experiments (*Figure 2C and E*). The pre-encounter mode has the highest threshold for eliciting a defensive response, which is only observed when *C. elegans* is exposed to life-threatening predators (*Figure 2F and G*). Notably, the pre-encounter defense mode reveals *C. elegans*'s ability to adapt its approach to new patches based on past experiences with patches inhabited by life-threatening predators, an adjustment that is reversible with subsequent exposure to predator-free patches (*Figure 1—figure supplement 3D*). This underscores the importance of threat severity in behavioral studies, contributing to the debate on the adequacy of using non-life-threatening stimuli to capture a full range of animal responses to danger.

Our study reveals that the defense modes in our nematode behavior model are not merely theoretical constructs but reflect physiologically distinct states driven by specific molecular mechanisms. While previous research has found that NLP-49 peptides largely act through SEB-3 in managing a variety of basal and stimulus-evoked stress responses (*Chew et al., 2018*), our findings indicate a divergence in how NLP-49 peptides and SEB-3 influence defensive behaviors, with this divergence varying across different defense modes. Specifically, altered *seb-3* but not *nlp-49* expression modulates behavior in the defense mode/phase with the highest predatory imminence (circa-strike, exit phase) (*Figure 3A and B*), while altered *nlp-49* but not *seb-3* expression influences behavior in the defense mode with the lowest predatory imminence (pre-encounter) (*Figure 3G and H*, *Figure 3—figure supplement 4*). Interestingly, while SEB-3 and NLP-49 peptides show some interdependence in the exit phase of the circa-strike mode (*Figure 4A*), NLP-49 peptides operate independently of SEB-3 in the pre-encounter mode (*Figure 4E and F*). Between these extremes of predatory imminence, changes in the expression of both *seb-3* and *nlp-49* affect defensive behaviors, but in ways that are incongruous or independent. While *nlp-49* and *seb-3* deletion mutants have similar enhanced phenotypes in the reentry phase of the circa-strike mode (*Figure 3C and D*), the loss of these phenotypes in double mutants (*Figure 4*) suggest that this behavior is not entirely modulated by NLP-49 peptides binding to the SEB-3 receptor. Although overexpression of *seb-3* and *nlp-49* both suppress post-encounter responses, *nlp-49* deletion enhances these responses while *seb-3* deletion has no effect (*Figure 3E and F*). Remarkably, *seb-3* overexpression animals exhibit normal behavior in the pre-encounter mode despite showing almost no response in the post-encounter mode, even though both involve extended exposure to predators (*Figure 3E and G*), illustrating that a deficit in one defense mode does not necessarily affect performance in another. This evidence highlights the physiological distinctiveness of the defense modes in our model system of nematode defensive behaviors. Furthermore, our results emphasize the complex interplay between SEB-3 and NLP-49 peptides, pointing to the need for further investigation into their underlying mechanisms. Moreover, our model serves as a useful instrument for an in-depth examination of the molecular signaling that drives defensive responses.

Our study extends previous findings on SEB-3's role in how *C. elegans* chooses between stimuli with opposite valences. In a prior study, male *C. elegans* were subjected to aversive blue light while mating, while researchers measured the time it took for males to disengage from mating in order to escape the blue light (*Jee et al., 2016*). This behavior most closely mirrors the exit phase of the circa-strike mode in our study, where *C. elegans* faces a choice between continuing to feed on a bacterial patch or exiting after a predator bite. In both cases, *C. elegans* must decide between pursuing a desirable activity (mating or feeding) and evading an unpleasant one (blue light or a predator). Our

research corroborates the other study's finding that enhancing SEB-3 function promotes *C. elegans* to persist in the appetitive behavior amidst aversive factors. Building on these findings, we delve into defensive behaviors shaped by repeated encounters with acute threats, aiming to understand the broader implications of molecular regulation in these scenarios. Future research could explore the response to other known paradigms for exposing *C. elegans* to natural threats like predatory fungi (*Maguire et al., 2011*) or artificial threats such as blue light or electric shocks (*Rankin et al., 1990*; *Tee et al., 2023*), shedding light on whether *C. elegans* differentiates between natural and artificial threats. Investigating roles of SEB-3 and NLP-49 peptides during extended exposure to mating under aversive conditions could provide further comparative insights, particularly on the generalizability of our study's conclusions across experimental conditions. It is important to clarify that our focus is on specific stress responses triggered by predatory threat, distinct from general stress indicators, such as hyperarousal and baseline locomotion. This distinction might explain why other studies linking SEB-3 and NLP-49 peptides to baseline stress behaviors have found contrasting results to ours regarding threat-induced responses (*Jee et al., 2013*; *Chew et al., 2018*), suggesting a need for further investigation to resolve these discrepancies.

Our study represents the continuation of ours and others' efforts to incorporate principles from ethology, behavioral ecology, and related fields into developing naturalistic and complex laboratory models of decision-making (*Krakauer et al., 2017*; *Mobbs et al., 2018*). Previously, we leveraged concepts from intraguild predation, neuroeconomics, and foraging theory to understand the motivations behind a predator's interactions with a prey that competes for the same bacterial food source (*Quach and Chalasani, 2022*). Using a similar foraging setup, the current study focuses on the prey's perspective and completes our exploration of both sides of this particular predator–prey interaction. Our work provides intricate and specific micro-scale insights into the behavioral ecology of flexible predator–prey interactions, which complements the more complex and broad insights of meso- and macro-scale ecology of predator–prey interactions in larger and less controlled ecosystems. Specifically, we address the concept of 'prey refuge' within the broader, more recent framework of the 'landscape of fear', coined in 2001 to describe spatial variation in prey perception of predation risk (*Laundré et al., 2001*). Our focused study on interactions between a single prey and a few predators contrasts with broader landscape of fear research, which often examines predator–prey dynamics of free-ranging predators and prey on complex landscapes, the cascading effects of these interactions on ecosystem structure, and how spatial variation in predation risk evolves over time (*Gaynor et al.,*

**Table 1.** *C. elegans* and *Pristionchus* spp. strains.

| Strain Name | Source | Genotype |
|---|---|---|
| N2 | CGC | Wildtype |
| RS5194 | *Click et al., 2009* | *P. pacificus* wild isolate |
| PS312 | *Click et al., 2009* | *P. pacificus* wild isolate |
| JU1051 | *Félix et al., 2013* | *P. uniformis* wild isolate |
| TU445 | *Ragsdale et al., 2013* | *P. pacificus eud-1(tu445)* |
| IV820 | This study | *seb-3(tm1848) X* outcrossed 4x |
| IV496 | This study | *seb-3(tm1848) X; ueEx309[Pseb-3::seb-3-GFP]* |
| AQ3644 | *Chew et al., 2018* | *nlp-49(gk546875)X* |
| AQ3853 | *Chew et al., 2018* | *nlp-49(gk546875)X; ljEx1004[Pnlp-49::Pnlp-49gDNA+ UTR::SL2-mKate2(25); unc-122::gfp(50)]* |
| AQ3701 | *Chew et al., 2018* | *seb-3(tm1848); nlp-49(gk546875)* |
| AQ3851 | *Chew et al., 2018* | *seb-3(tm1848); ljEx1004[Pnlp-49::Pnlp-49 gDNA+ UTR::SL2-mKate2(25); unc-122::GFP(50)]* |

Strains are *C. elegans* unless otherwise indicated.
CGC, Caenorhabditis Genetics Center.

*2019*; *Palmer et al., 2022*). A common challenge in landscape of fear studies is reconciling actual predation risk with perceived predation risk. Accurately predicting the impact of prey's anti-predator behaviors on population and ecosystem levels necessitates a deep understanding of the external and internal factors influencing prey responses at the individual level. While models of prey refuge have laid the groundwork for exploring the landscape of fear in more complex ecological systems (*Sih, 1987*), our work adds a new dimension by considering predator imminence as another critical factor influencing prey's spatial behavior. This study, together with its companion study on predator decision-making (*Quach and Chalasani, 2022*), demonstrates that complex behavioral theories applicable to advanced nervous systems are also relevant to the simpler neural circuits of nematodes. By deconstructing complex behaviors and decision-making relevant to a nematode's natural life, we can adapt existing theories to the unique aspects of nematode life and interactions.

## Materials and methods

### *Caenorhabditis elegans* and *Pristionchus* spp. strains

The nematode strains used in this study are shown in *Table 1*.

### Nematode culture and selection for behavioral experiments

*Caenorhabditis elegans* and *Pristionchus* spp. animals were cultured using standard methods (*Stiernagle, 2006*). Day 1 adult hermaphrodite *C. elegans* were used for all behavioral experiments. For the hermaphroditic *P. pacificus* strains (TU445, PS312, RS5194), day 1 hermaphrodites were used. For the gonochoristic *P. uniformis* strain (JU1051), we used day 1 females as they are the similar in size and morphology to *P. pacificus*. Additionally, JU1051 females were used to avoid attempts by male JU1051 to mate with *C. elegans* hermaphrodites.

### Behavioral imaging

Behavioral images and video recordings were acquired using an optiMOS sCMOS camera (QImaging) and Streampix software. To keep animals within field-of-view, corrals were made by using a hole punch or a die-cut machine (Cricut Maker 3) to cut 6 mil transparent mylar sheets into desired arena configurations.

### Bacterial patches

To create stocks of bacterial liquid cultures, lysogeny broth (LB) was inoculated with a single colony of *E. coli* OP50, grown at room temperature overnight, and then stored at 4°C for up to 2 months. To produce a working liquid culture, the stock liquid culture was diluted with LB to an $OD_{600}$ value of 0.018 (standard density) or 0.06 (high density) using a NanoDrop spectrophotometer. Bacterial patches were created by dispensing 0.3 μl of cold working liquid culture onto cold 3% agar NGM plates (*Stiernagle, 2006*), resulting in patches that are approximately 2 mm in diameter. Bacterial patches were grown for 24 hours at 20°C. Fully grown patches were stored at 4°C and allowed to come to room temperature for 1 hour before use in behavioral experiments. All bacterial patches were inspected for roundness and size. Standard patches were characterized by a sharp raised boundary, while high-density lawns exhibited a thick, wide boundary that transitioned smoothly into the interior of the patch.

### Circa-strike behaviors

To provide ample space for *C. elegans* to leave and avoid the bacterial patch, a 2 mm bacterial patch was centered inside a 9.5 mm diameter circular arena. The resulting arena allowed *C. elegans* to leave the bacterial patch from any part of the patch boundary, and the bacteria-free ring surrounding the bacterial patch was over three body lengths wide. 1 × *C. elegans* adult and 4 × *Pristionchus* spp. adults (or no predators) were placed in the arena and recorded for 1 hour. Video recordings were manually scored for timestamps when *C. elegans*: (1) exhibits an escape response to a bite, (2) exits the bacterial patch, or (3) reenters the bacterial patch. Scoring criteria for bites were previously described in *Quach and Chalasani, 2022*. A bite-induced escape response was defined as *C. elegans* rapidly accelerating away from the bite (*Pirri and Alkema, 2012*). An exit was defined as *C. elegans* transitioning from being inside the bacterial patch to moving its body completely outside of the patch. A reentry was defined as *C. elegans* transitioning from being completely outside the bacterial patch

to being partially (with head) or completely inside the bacterial patch. If *C. elegans* was visibly dead or injured as indicated by abnormal locomotion, the remainder of the video was excluded. Exit latency was measured as the time between a bite and the point at which the *C. elegans* head enters the off-patch area, for events in which *C. elegans* fully exits the patch. The probability of a *C. elegans* individual leaving the bacterial patch after escaping a bite, P(exit|escape), was calculated as the number of exits divided by the number of bite-induced escapes responses. Reentry latency was calculated as the time between the point at which the *C. elegans* head enters the off-patch area and the point at which the *C. elegans* head enters the patch, for events in which *C. elegans* fully exits the patch.

## Post-encounter behaviors

The arena setup and rationale were the same as for assessing patch leaving (see above section). 1 × *C. elegans* adult and 4 × *Pristionchus* spp. adults (or no predators) were placed in the arena for 6 hours. *C. elegans* was visually assessed every hour for whether it was fully inside the bacterial patch or in a stable outstretched feeding posture. An outstretched feeding posture was defined as *C. elegans* having only its head inside the patch or feeding on a bacterial trail outside of the patch, with the rest of its body outside the bacterial patch and stretched out from its typical sinusoidal waveform. To ensure accurate assessment of feeding posture choice rather than location at a point in time, we wait up to 10 minutes for the first persistent feeding posture (stationary for >10 seconds) if *C. elegans* is in transition between on and off-patch states. Any time points without stable inside-patch or outstretched feeding postures were excluded from analysis. Dead or injured *C. elegans* were also excluded.

## Pre-encounter behaviors

Two different arena setups were used: a wide exit arena and a narrow exit arena. The wide exit arena setup and rationale were the same as for assessing patch leaving and outstretched feeding posture (see above sections). To create a narrow exit arena, a two-chamber arena was designed such that a pair of 2 mm diameter circular cutouts were connected by a 3 mm × 0.7 mm rectangular cutout, resulting in a dumbbell shape (*Figure 1—figure supplement 1C*). A 2 mm bacterial patch was centered inside one of the 2 mm diameter circular cutouts, such that the patch perimeter was entirely surrounded by the corral except for a 0.7 mm opening. While the wide exit arena allowed *C. elegans* to exit and enter the bacterial patch anywhere along the patch circumference, the narrow exit arena allowed exit and entry to only 1/9 of the circumference. For the predator exposure phase, 1 × *C. elegans* adult and 4 *Pristionchus* spp. adults (or no predators for mock exposure) were placed in the arena for 4 hours (unless otherwise stated). After the predator exposure phase, *C. elegans* was assessed for normal and vigorous locomotion. We especially check for the typical sinusoid waveform of its body as it crawls on non-bacterial surfaces as injury to any part of the body can disrupt the sinusoid waveform. *C. elegans* individuals were excluded if they were visibly dead, paralyzed, or injured as indicated by abnormal locomotion. In particular, we looked for the vigorous movement and sinusoid waveform of typical locomotion. In the predator-free phase, *C. elegans* was transferred to a new arena that was identical to the one used for predator or mock exposure, but without predators present. *C. elegans* was placed in the bacteria-free circular cutout of the arena, and patch exploration began once *C. elegans* touched its nose to the predator-free patch. After 15 minutes of patch exploration, an image of the bacterial patch was taken. Entry latency was measured as the time between *C. elegans* being placed in the arena to the point at which the *C. elegans* head contacts the patch. Patch exploration was measured as the number of bins containing worm tracks in the image of the bacterial patch. To count the number of bins containing worm tracks, a 10 × 10 square grid was superimposed on top of the bacterial patch image in MATLAB (*Figure 1—figure supplement 1*). The body length of *C. elegans* is about 5 bins wide, while the portion of the head that can move while the rest of the body is stationary is about 1 bin wide. In extinction experiments, *C. elegans* was transferred to a new predator-free arena every hour for 6 hours following the predator exposure phase.

## Baseline on-bacteria escape speed

To maximize predator–prey encounter frequency and limit locomotion to only one kind of surface, a 2 mm bacterial patch was centered inside a 2 mm diameter circular arena. The resulting arena was completely filled with bacteria and lacked any bacteria-free agar surface where *C. elegans* can escape to and move quickly. 1 × *C. elegans* adult and 4 × *P. pacificus* adults were placed in the arena and

recorded. Video recordings were manually scored for bite event start times. If *C. elegans* was visibly dead or injured as indicated by abnormal locomotion, the remainder of the video was excluded. The *C. elegans* nose was manually tracked in MATLAB throughout the escape window, defined as the first 15 seconds immediately after being bitten. The escape speed following each bite was calculated as the distance traveled the escape window, divided by the duration of the escape window or the interval between two bites if one occurs within 15 seconds.

## Baseline off-bacteria, predator-free speed

To best replicate conditions of the off-patch circa-strike environment, we used the same 9.5 mm diameter circular arena but omitted bacteria and predators. A single *C. elegans* adult was placed into the center of the arena and recorded for 5 minutes. Video recordings were downsampled to 3 fps and manually tracked in MATLAB to obtain head locations. The average speed was calculated as the distance traveled divided by 5 minutes.

## Statistical methods

Statistical test parameters and outcomes are indicated in figure legends.

Sample size was determined by power analysis of pilot experiments. Animals were pseudorandomized to balance conditions. Images and videos were named with non-descriptive identifiers so that the analyses were blind to condition. All data describe biological replicates. Exclusion criteria described above were pre-established for omitting data that would be inherently uninformative for the intended purpose of the metric of interest. Due to predator-induced injury or death, omission of data was due to attrition.

For datasets with nominal independent variables and measurement dependent variables, assumptions for statistical tests were evaluated prior to select an appropriate parametric or non-parametric test for comparing groups. The Shapiro–Wilk test was used to test for normality within each group, while Levene's test was used to test for homogeneity of variances across groups. For comparisons between two groups, Student's *t*-test was used to compare normally distributed groups with equal variances, Welch's *t*-test was used to compare normally distributed groups with unequal variances, and Wilcoxon rank sum test was used to compare non-normally distributed groups. For paired comparisons, the paired *t*-test was used to compare groups with normally distributed differences. For comparisons between more than two groups, one-way ANOVA with Tukey's post hoc test was used for normally distributed groups with equal variances, Welch's ANOVA with Games–Howell post hoc test was used for normally distributed groups with unequal variances, and Kruskal–Wallis test with Dunn's post hoc test was used for non-normally distributed groups. To adjust p-value for multiple comparisons between independent comparisons, we used the Benjamini–Hochberg method. To avoid making assumptions of normality in error bar representation, we performed non-parametric bootstrap resampling ($1 \times 10^3$ iterations) to obtain empirical 95% CIs containing the mean.

For datasets in which both independent and dependent variables are nominal, we used Fisher's exact test. To adjust p-value for multiple comparisons, we used the Benjamini–Hochberg method.

For datasets in which both independent and dependent variables are continuous measurements, we represented the data as a 2-D plot and calculated linear regression lines with shaded regions representing 95% CIs from linear regression models. To compare two linear regression lines, we used the Kruskal–Wallis test on the residuals of linear regression models.

For datasets in which the dependent variable is a measurement that varies over 'time' (i.e., consecutive bites), we used non-parametric bootstrap resampling with replacement for $1 \times 10^5$ iterations to obtain empirical 95% CIs. Timecourses were compared by identifying areas of non-overlap as statistically significant ($p < 0.05$).

All statistical analyses were carried out with the R statistical software (*R Development Core Team, 2017*). The additional package `multcomp` was used to conduct linear hypotheses with single-step adjustment for multiple comparisons (*Hothorn et al., 2008*). The additional package `boot` was used to perform non-parametric bootstrap resampling to obtain empirical 95% CIs containing the mean (*Canty, 2017*). The additional package `rstatix` was used to perform the Games–Howell test. The additional package FSA was used to perform Dunn's test.

## Materials availability
Newly generated animal strains are available upon request.

## Acknowledgements
We would like to thank the Caenorhabditis Genetics Center (United States), National BioResource Project (Japan), WR Schafer, RL Hong, C Jee, and RJ Sommer for providing nematode strains. We would also like to thank Amy Pribadi and Kirthi Reddy for their help outcrossing strains, as well as Jess Haley for her comments on the manuscript. This work was supported by an NIH R01 grant (R01 MH113905), the Maximillian E and Marion O Hoffman Foundation (KTQ), and the Paul F Glenn Center for Biology of Aging Research (KTQ).

## Additional information

### Funding

| Funder | Grant reference number | Author |
| --- | --- | --- |
| Paul F. Glenn Center for Biology of Aging Research | | Kathleen T Quach |
| Maximillian E. and Marion O. Hoffman Foundation | | Kathleen T Quach |
| National Institute of Mental Health | R01MH113905 | Sreekanth H Chalasani |

The funders had no role in study design, data collection and interpretation, or the decision to submit the work for publication.

### Author contributions
Kathleen T Quach, Conceptualization, Data curation, Software, Formal analysis, Funding acquisition, Investigation, Visualization, Methodology, Writing – original draft, Project administration, Writing – review and editing; Gillian A Hughes, Data curation, Formal analysis, Investigation, Writing – review and editing; Sreekanth H Chalasani, Supervision, Funding acquisition, Project administration, Writing – review and editing

### Author ORCIDs
Kathleen T Quach [ORCID] https://orcid.org/0000-0001-8498-1464
Sreekanth H Chalasani [ORCID] https://orcid.org/0000-0003-2522-8338

Reviewer #1 (Public review): https://doi.org/10.7554/eLife.98262.2.sa1
Reviewer #2 (Public review): https://doi.org/10.7554/eLife.98262.2.sa2
Author response https://doi.org/10.7554/eLife.98262.2.sa3

## Additional files

### Supplementary files
MDAR checklist

### Data availability
All data generated or analysed during this study are included in the manuscript and supporting files.

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
