## [Editor Report · eLife assessment]

This study presents a **valuable** finding on predator threat detection in *C. elegans* and the role of neuropeptide systems in defensive behavioral strategies. The evidence supporting the conclusions is **solid**, although additional analyses and control experiments would strengthen the claims of the study. Overall, the work is of interest to the *C. elegans* community as well as neuroethologists and ecologists studying predator-prey interactions.

---

## [Referee Report · Reviewer #1 (Public review)]

Summary:

In this manuscript, Quach et al. report a detailed investigation into the defense mechanisms of *Caenorhabditis elegans* in response to predatory threats from Pristionchus pacificus. Based on principles from predatory imminence and prey refuge theories, the authors delineate three defense modes (pre-encounter, post-encounter, and circa-strike) corresponding to increasing levels of threat proximity. These modes are observed in a controlled but naturalistic setup and are quantified by multiple behavioral outputs defined in time and/or space domains allowing nuanced phenotypic assays. The authors demonstrate that *C. elegans* displays graded defense behavioral responses toward varied lethality of threats and that only life-threatening predators trigger all three defense modes. The study also offers a narrative on the behavioral strategies and underlying molecular regulation, focusing on the roles of SEB-3 receptors and NLP-49 peptides in mediating responses in these defense modes. They found that the interplay between SEB-3 and NLP-49 peptides appears complex, as evidenced by the diverse outcomes when either or both genes are manipulated in various behavioral modes.

Strengths:

The paper presents an interesting story, with carefully designed experiments and necessary controls, and novel findings and implications about predator-induced defensive behaviors and underlying molecular regulation in this important model organism. The design of experiments and description of findings are easy to follow and well-motivated. The findings contribute to our understanding of stress response systems and offer broader implications for neuroethological studies across species.

Weaknesses:

Although overall the study is well designed and movitated, the paper could benefit from further improvements on some of the methods descriptions and experiment interpretations.

---

## [Referee Report · Reviewer #2 (Public review)]

In this study, the authors characterize the defensive responses of *C. elegans* to the predatory Pristionchus species. Drawing parallels to ecological models of predatory imminence and prey refuge theory, they outline various behaviors exhibited by *C. elegans* when faced with predator threats. They also find that these behaviors can be modulated by the peptide NLP-49 and its receptor SEB-3 in various degrees.

The conclusions of this paper are mostly well-supported, the writing and the figures are clear and easy to interpret. However, some of the claims need to be better supported and the unique findings of this work should be clarified better in text.

(1) Previous work by the group (Quach, 2022) showed that Pristionchus adopt a "patrolling strategy" on a lawn with adult *C. elegans* and this depends on bacterial lawn thickness. Consequently, it may be hypothesized that *C. elegans* themselves will adopt different predator avoidance strategies depending on predator tactics differing due to lawn variations. The authors have not shown why they selected a particular size and density of bacterial lawn for the experiments in this paper, and should run control experiments with thinner and denser lawns with differing edge densities to make broad arguments about predator avoidance strategies for *C. elegans*. In addition, *C. elegans* leaving behavior from bacterial lawns (without predators) are also heavily dependent on density of bacteria, especially at the edges where it affects oxygen gradients (Bendesky, 2011), and might alter the baseline leaving rates irrespective of predation threats. The authors also do not mention if all strains or conditions in each figure panel were run as day-matched controls. Given that bacterial densities and ambient conditions can affect *C. elegans* behavior, especially that of lawn-leaving, it is important to run day-matched controls.

(2) Both the patch-leaving and feeding in outstretched posture behaviors described here in this study were reported in an earlier paper by the same group (Quach, 2022) as mentioned by the authors in the first section of the results. While they do characterize these further in this study, these are not novel findings of this work.

(3) For Figures 1F-H, given that animals can reside on the lawn edges as well as the center, bins explored are not a definitive metric of exploration since the animals can decide to patrol the lawn boundary (especially since the lawns have thick edges). The authors should also quantify tracks along the edge from videographic evidence as they have done previously in Figure 5 of Quach, 2022 to get a total measure of distance explored.

(4) Where were the animals placed in the wide-arena predator-free patch post encounter? It is mentioned that the animal was placed at the center of the arena in lines 220-221. While this makes sense for the narrow-arena, it is unclear how far from the patch animals were positioned for the wide exit arena. Is it the same distance away as the distance of the patch from the center of the narrow exit arena? Please make this clear in the text or in the methods.

(5) Do exit decisions from the bacterial patch scale with number of bites or is one bite sufficient? Do all bites lead to bite-induced aversive response? This would be important to quantify especially if contextualizing to predatory imminence.

(6) Why are the threats posed by aversive but non-lethal JU1051 and lethal PS312 evaluated similarly? Did the authors characterize if the number of bites are different for these strains? Can the authors speculate on why this would happen in the discussion?

(7) The authors indicate that bites from the non-aversive TU445 led to a low number of exits and thus it was consequently excluded from further analysis. If anything, this strain would have provided a good negative control and baseline metrics for other circa-strike and post-encounter behaviors.

1. For Figures 3 G and H, the reduction in bins explored (bins_none - bins_RS1594) due to the presence of predators should be compared between wildtype and mutants, instead of the difference between none and RS5194 for each strain.

(9) While the authors argue that baseline speeds of seb-3 are similar to wild type (Figure S3), previous work (Jee, 2012) has shown that seb-3 not only affects speed but also roaming/dwelling states which will significantly affect the exploration metric (bins explored) which the authors use in Figs 3G-H and 4E-F. Control experiments are necessary to avoid this conundrum. Authors should either visualize and quantify tracks (as suggested in 3) or quantify roaming-dwelling in the seb-3 animals in the absence of predator threat.

(10) While it might be beyond the scope of the study, it would be nice if the authors could speculate on potential sites of actions of NLP-49 in the discussion, especially since it is expressed in a distinct group of neurons.

---

## [Author Response]

We are proceeding without revisions as the first author has chosen to withdraw from the project and will not be contributing further.